# Promoting Judicious Antimicrobial Use in Beef Production: The Role of Quarantine

**DOI:** 10.3390/ani12010116

**Published:** 2022-01-04

**Authors:** Matteo Santinello, Alessia Diana, Massimo De Marchi, Federico Scali, Luigi Bertocchi, Valentina Lorenzi, Giovanni Loris Alborali, Mauro Penasa

**Affiliations:** 1Department of Agronomy, Food, Natural Resources, Animals and Environment (DAFNAE), University of Padova, 35020 Legnaro, VEN, Italy; alessiadiana84@gmail.com (A.D.); massimo.demarchi@unipd.it (M.D.M.); mauro.penasa@unipd.it (M.P.); 2Sector Diagnostic and Animal Health, Istituto Zooprofilattico Sperimentale della Lombardia e dell’Emilia Romagna ‘Bruno Ubertini’ (IZSLER), 25124 Brescia, LOM, Italy; federico.scali@yahoo.it (F.S.); giovanni.alborali@izsler.it (G.L.A.); 3Italian National Reference Center for Animal Welfare (CReNBA), Istituto Zooprofilattico Sperimentale della Lombardia e dell’Emilia Romagna ‘Bruno Ubertini’ (IZSLER), 25124 Brescia, LOM, Italy; luigi.bertocchi@izsler.it (L.B.); valentina.lorenzi@izsler.it (V.L.)

**Keywords:** antimicrobial resistance, antimicrobial stewardship, biosecurity, beef cattle, treatment incidence

## Abstract

**Simple Summary:**

The development of cost-effective strategies that can be easily implemented on-farms is pivotal to promote a more judicious use of antimicrobials and its reduction in livestock industry. Indeed, inappropriate use of antimicrobials is linked to the phenomenon of antimicrobial resistance, a global health concern for both humans and animals. Studies on other food-producing species have confirmed the effectiveness of biosecurity measures on the reduction of antimicrobials, of while little is still known in beef production. Thus, this study aimed to investigate the effect of quarantine as a strategy to reduce medications in beef production. This measure resulted to be a viable strategy to reduce antimicrobials in beef cattle without compromising animal health and performance. The reduction was evident especially with regards to treatments administered for respiratory diseases, indeed the bovine respiratory disease is one of the most detrimental health issues affecting beef cattle. Penicillins was the most used class of antimicrobials, highlighting the need for an urgent decrease of such broad-spectrum medications, known for their contribution to the development of resistance. Although implementing new strategies on-farm can be costly for farmers, the reduction of antimicrobials on the long term and the support from EU authority may help to overcome some initial disadvantage.

**Abstract:**

Judicious antimicrobial stewardship in livestock industry is needed to reduce the use of antimicrobials (AMU) and the associated risk of antimicrobial resistance. Biosecurity measures are acknowledged for their role against the spread of diseases and the importance in reducing AMU in different species. However, their effectiveness in beef production has been scarcely considered. The aim of this study was to investigate the effect of the quarantine strategy on AMU in beef cattle. A total of 1206 Charolaise animals in five farms were included in the trial. Roughly half of the animals followed the standard procedure of the fattening cycle (no-quarantine; NO-QUA group) and half followed a 30-day period of quarantine (QUA group) since their arrival. Performance and antimicrobial data were recorded and a treatment incidence 100 (TI100it) per animal was calculated. Penicillins was the most used class of antimicrobials. Differences between groups were significant for males only, with NO-QUA group having greater TI100it (3.76 vs. 3.24; *p* < 0.05) and lower body weight at slaughter (713.4 vs. 723.7 kg; *p* < 0.05) than QUA group. Results suggest that quarantine strategy can reduce AMU in males without compromising their performance, whereas further investigation is needed for females.

## 1. Introduction

The importance of antimicrobials in animal production is well-known. Since their discovery in the late 1940s, they have been essential to tackle infectious diseases, especially in intensive farming systems where pathogens are more likely to arise [1,2]. However, the misuse of antimicrobials and the associated risk of antimicrobial resistance (AMR) are global issues that jeopardize both human and animal health alike [3,4]. Animal production is a major contributor to the increase of AMR [3,4,5] thus highlighting the need of addressing antimicrobial use (AMU) in animal food-producing sectors. For example, some studies showed that an overuse of antimicrobials can select for the carriage of resistant bacteria in beef production [6,7,8]. Current research worldwide is mainly focused on the development of cost-effective strategies to be easily implemented in livestock farms and to contribute to the reduction of AMU. Alternatively, management strategies reported in the literature for the livestock species include supplementation of diet with additives (e.g., essential oils and clay minerals), organic acids or probiotics [9,10,11], targeted vaccinations, and changes in husbandry practices [10,12]. These strategies will be pivotal to promote a more judicious AMU. To achieve this goal the collection of accurate data on AMU at farm level is crucial [13]. One of the most recognized indicators to estimate AMU is the treatment incidence 100 [14,15] which is calculated through the defined daily dose animal (DDDA) established at both EU and Italian level.

Biosecurity is also important to reduce AMU [12,16] because it accounts for measures which may contribute to prevent the introduction and spread of diseases in the herd [17]. Effective biosecurity measures to monitor and reduce the risk of diseases, such as the bovine respiratory disease (BRD), have been identified in beef production [18,19], yet their implementation is still scarce due to required investments and lack of information on their efficacy [20]. Some examples of biosecurity measures are the separation of younger calves from older cattle, the reduction of the stocking rate, the application of a quarantine period to the imported animals, the testing and culling of clinical suspects only, and the schedule of vaccination programs [10,21]. Unlike for beef production, studies on other livestock species have already confirmed the positive link between biosecurity and animal health, welfare, and productivity as well as the effectiveness of these measures on AMU [22]. For instance, Postma et al. [12] reported the efficacy of improving internal and external biosecurity measures to reduce AMU in pigs without compromising animal performances.

The effectiveness of biosecurity in the reduction of AMU would encourage an antimicrobial stewardship in beef production. So far, management indicators recognized as contributing factors significantly associated with AMU in beef cattle have been investigated mainly through observational studies [23,24]. For instance, a study on Swiss veal farms observed a significant association between lack of quarantine upon arrival at the fattening farm and increase of treatment incidence [24]. In addition, the duration of the fattening period, the quarantine, and the space of feeding areas were significantly associated with AMU in veal calves [23]. A recent study carried out on Swiss veal farms evaluated a novel outdoor concept for calf fattening, which resulted in a drastic reduction of AMU without compromising animal health [25]. To the best of our knowledge, little is known about cost-effective strategies that may promote a prudent antimicrobial stewardship in beef production. Therefore, the aim of the present study was to investigate the effect of quarantine of imported beef cattle as a strategy to reduce AMU on-farm.

## 2. Materials and Methods

This study was approved by the Ethical Committee for the Care and Use of Experimental Animals of the University of Padova, Italy (approval no. 74/2018) and was conducted in accordance with Italian law (Decreto legislativo no. 26/2014) and EU Directive 2010/63/EU on the protection of animals used for scientific purposes.

### 2.1. Specialized Fattening System in Italian Beef Cattle

Approximately 70% of beef cattle produced in Italy are reared in specialized fattening farms in the north-east of the country and around 90% of young animals farmed in this area are imported from France [26]. In France, animals are reared at pasture until 10–14 months of age, and then transferred to specific collection centers and mixed with animals of other farms located in different French departments to create homogeneous batches according to body weight (BW), breed, and sex. Within 2–4 days, animals are purchased by Italian beef fatteners and transported to Italy. The intensive conditions under which animals are fattened in Italy allow them to reach the slaughter weight after 6–7 months from arrival. The diet supplied to the animals consists of a total mixed ration with high proportion of concentrates, different proportions of feedstuffs according to breed, sex and fattening stage, and mineral and vitamin supplementations. The typical housing system of Italian beef farms, which generally has a turnover of about two fattening cycles per year, consists of closed or open barns with multiple pens. Each pen has fully slatted or concrete floors with straw bedding [27].

### 2.2. Experimental Design and Treatment Groups

A total of 1206 Charolaise cattle (576 males and 630 females) entered five commercial specialized beef fattening farms associated to a cooperative of beef producers (AZoVe) located in Veneto region (Italy) between July 2018 and August 2019. Three farms reared only females and two only males, and all of them purchased animals from France in five different periods between July 2018 and August 2019 with an average of 240 animals per period. For each period, only one truck with animals of the same sex and homogeneous BW arrived to each of the five farms. At their arrival to the farm, animals were weighed and divided in two experimental groups, namely quarantine (QUA) and no-quarantine (NO-QUA), which were allocated in two different buildings of the farm. Two pens per experimental group were available for the trial and all pens were balanced for kg of initial BW per m^2^. On average, each pen contained 12 animals and the surface available per animal averaged 5.3 m^2^. Animals allocated to the NO-QUA group followed the standard fattening cycle and thus since the beginning of the trial joined animals that were already present in the farm, i.e., those not included in the trial but fattened in the same building of NO-QUA animals. Instead, animals allocated to the QUA group followed a 30-day period of quarantine since their arrival to the farm before moving to the same building of NO-QUA group. Prior to the allocation of the animals, the building designated for QUA animals was cleaned and sanitized. Both QUA and NO-QUA groups started the fattening cycle on the same day. The animals went through the same health protocol. Specifically, they did not receive any vaccines nor antimicrobial treatments in France. Whereas, at their arrival to the Italian fattening farms, the same vaccination program (i.e., polyvalent vaccine for BRD) and parasitic control program were administered to the animals.

In total, 578 animals (48%) from the five periods were assigned to the NO-QUA group (264 males, 314 females) and 628 animals (52%) to the QUA group (316 males, 312 females). The diet provided to the animals was the same for QUA and NO-QUA groups but differed according to sex (Table 1). Since it was necessary to comply with the needs of the farmers and the routine management procedures of their farms, a non-randomized controlled intervention study was considered the most suitable approach. Animals of a truck were a mix of different French farms and this, together with the creation of pens with similar kg of BW per m^2^, contributed to minimize the potential bias related to the allocation of the animals to the two experimental groups. In fact, for male-rearing farms, animals allocated to the QUA group were purchased from 12 departments and 138 farms of origin while animals of NO-QUA group were purchased from 10 French departments and 108 farms of origin. For female-rearing farms, animals were originally purchased from 18 French departments and 156 farms of origin for QUA group and 21 departments and 162 farms of origin for NO-QUA group.

Information on date of birth, date of start and end of the fattening cycle (the mean duration of the fattening cycle was 193 days), BW at arrival to the fattening farm (BW0), BW at 30 days after arrival to the fattening farm (BW30), and BW at the end of the fattening cycle (BWfinal) were collected for each animal. In particular, animals were individually weighed over three time points: (1) once arrived at the farm prior to their allocation to the experimental groups, (2) after 30 days since their arrival to the farm, and (3) prior to transport to the slaughterhouse. Data on BW were used to calculate the average daily gain during the first 30 days from arrival to the fattening farm (ADG30, kg/d) and at the end of the fattening cycle (ADGtot, kg/d). Reasons of culling during the fattening cycle (e.g., injury, death) were also recorded. The afore-mentioned variables were also used to obtain the season of arrival at the fattening farm, the length of the fattening cycle (days), the number of deaths and the mortality rate (%).

### 2.3. Quantification of Antimicrobial Use

Data on the number of parenteral treatments administered to the animals, the date of treatment, the reason of administration and the amount (mL) of antimicrobial used per each parenteral treatment were recorded throughout the fattening cycle. Antimicrobials were administered by veterinarians employed by the AZoVe cooperative of beef producers. They are all equally trained and stay blind on whether the farms are or are not involved in research trials. Thirteen veterinary medicinal products (VMP) containing antimicrobials were used in the studied farms. A defined daily dose animal for Italy (DDDAit) was assigned to each active ingredient (AI) with antimicrobial activity of those VMP. A DDDAit represents the dose (mg) of the AI administered per kg of BW per day and it was established during the development of the ClassyFarm integrated monitoring system (www.classyfarm.it) of the Italian Ministry of Health. In order to quantify the frequency of treatment, which allows for a better monitoring of AMU [14,15], an index called treatment incidence 100 for Italy (TI100it) was calculated per each VMP [15] at animal level using the following formula, modified from Timmerman et al. (2006) Equation (1):TI100it = amount of AI administered per animal (mg)/[DDDAit (mg/kg/day) × standard body weight (kg) × standard days at risk] × 100(1)
where ‘standard body weight’ is the average expected BW of the animal at treatment (400 kg) and ‘days at risk’ the standard number of days of the fattening cycle (230 days). The TI100it of all VMP were summed up to obtain a total TI100it per animal and considering all antimicrobial administrations carried out during the whole fattening cycle for both groups. If a VMP had two AI, both were considered in the calculation of the DDDAit as two different treatments.

### 2.4. Statistical Analysis

Data were analyzed using the software SAS 9.4 (SAS Institute Inc., Cary, NC, USA). Animal was the experimental unit. Data were tested for normality. Males and females were analyzed separately because none of the farms had both sexes. Descriptive statistics of BW0, BW30, BWfinal, length of the fattening cycle, ADG30, ADGtot, and TI100it per sex were calculated. In addition, mortality rate, number of animals treated per sex, number and percentage of parenteral treatments per class of antimicrobials, percentage of parenteral treatments according to the reason of administration and number and percentage of parenteral treatments per experimental group per sex (QUA and NO-QUA for males, QUA and NO-QUA for females) were calculated. A Chi-Square test was performed to check for differences regarding the reasons of treatment between groups (QUA and NO-QUA).

To investigate the effect of the strategy of quarantine on animal performance, two ANOVA tests (one for male-rearing farms and one for female-rearing farms) were performed using the GLM procedure of SAS. BW0 was transformed into a categorical variable through the creation of three classes of BW (low, medium, high) within sex according to mean ± 0.5 SD. The following linear model was used Equation (2):y*_ijkl_* = µ + farm*_i_* + quarantine*_j_* + season*_k_* + iBW*_l_* + (farm × quarantine)*_ij_* + (quarantine × season)*_ik_* + e*_ijkl_*(2)
where µ is the overall intercept of the model; y*_ijkl_* is the dependent variable (BW30, BWfinal, ADGtot or ADG30); farm*_i_* is the fixed effect of the *_i_*th farm (two farms for males and three for females); quarantine*_j_* is the fixed effect of the *_j_*th experimental group (NO-QUA and QUA); season*_k_* is the fixed effect of the *_k_*th season of arrival of the animal to the fattening farm (spring: March, April, May; summer: June, July, August; autumn: September, October, November; winter: December, January, February); iBW*_l_* is the fixed effect of the *_l_*th class of BW of the animal at arrival to the fattening farm (low, medium, high); (farm × quarantine)*_ij_* is the fixed interaction effect between farm and experimental group; (quarantine × season)*_jk_* is the fixed interaction effect between experimental group and season of arrival of the animal to the fattening farm; and e*_ijkl_* is the random residual. Data are presented as least squares means ± standard error. Multiple comparisons among least squares means of the fixed effects were performed through Bonferroni post-hoc test. The criterion for statistical significance was established at *p* < 0.05 and for statistical trend at 0.05 < *p* < 0.10.

A third model was built to investigate the effect of the strategy of quarantine on AMU. Only males were included in the statistical analysis as the number of females treated with antimicrobials was very low. Data were not normally distributed, thus the TI100it was analyzed using a generalized linear mixed model with gamma distribution and log link function in GLIMMIX procedure of SAS. Before performing the analysis, a constant of 3 was added to TI100it in order to avoid an over-estimation of the index during the statistical analysis due to the high number of zeros (animals not treated) and the impossibility of modelling the log. The model included farm, quarantine, and season of arrival as categorical fixed effects, BW0 as linear covariate and intercepts of animal ID nested within period of purchasing as random effect. Goodness of fit of the model was evaluated by checking Akaike’s Information Criterion and Bayesian Information Criterion of each step of model building. Results are presented as least squares means ± standard error. Tukey–Kramer post hoc adjustment was used for multiple comparisons of least squares means of the fixed effects. The criterion for statistical significance was set at *p* < 0.05 and for statistical trend at 0.05 < *p* < 0.10.

## 3. Results

### 3.1. Descriptive Statistics of Performance Traits

There was a large variability on performance traits between farms rearing different sexes. In particular, male-rearing farms had heavier animals than female-rearing farms, either at the start (403.4 vs. 320.6 kg, respectively) and end of the fattening cycle (719.6 vs. 559.6 kg; Table 2). The mortality of QUA and NO-QUA animals was 0.96% and 1.04%, respectively.

### 3.2. Effects of Farm, Quarantine, and Season on Performance Traits

The two male-rearing farms differed significantly for all performance traits, except for ADG30. Quarantine group males had higher BW30, BWfinal, ADG30 and ADGtot than NO-QUA males (*p* < 0.05; Table 3). There was also an effect of the season of arrival to the fattening farm on all performance traits. In particular, animals that entered the fattening farm in spring had significantly higher BW30, BWfinal, ADG30, and ADGtot compared with animals that entered the farm in other seasons (*p* < 0.05; Table 3).

Across the three female-rearing farms, there were significant differences for BW30, BWfinal and ADG30 (*p* < 0.05; Table 4). No differences were reported between QUA and NO-QUA groups (*p* > 0.05), whereas season of arrival affected performance traits. Specifically, females that entered the farm in winter had significantly higher BWfinal and ADGtot than females that entered the farm in other seasons (*p* < 0.05; Table 4). Least squares means for the interaction between farm x quarantine and quarantine x season were not significant, thus no results were provided.

### 3.3. Descriptive Statistics of Antimicrobial Use

One-hundred-and-fifty-six animals out of 1206 received at least one treatment during the fattening cycle. Specifically, 126 out of 156 were males and 30 females. A total of 675 parenteral treatments were administered during the trial: 57.0% were administered to NO-QUA animals and 43.0% to QUA animals. Three of the 13 VMP used in the farms were composed of two AI belonging to different classes of antimicrobials. Thus, we also calculated the number and percentage of parenteral treatments administered to the animals identifying each treatment with the number of AI of each VMP. If each AI of a VMP is counted as a separate treatment, the number of parenteral treatments increased to 763 (Table 5). In general, penicillins were the most frequently used antimicrobials (29.2%) followed by amphenicols (19.7%), fluoroquinolones (15.7%), and aminopenicillins (13.9%; Table 5). The TI100it averaged 0.76 ± 2.65 for males (range: 0 to 36.02) and 0.13 ± 0.69 for females (range: 0 to 9.17).

Overall, the main reasons of administration of antimicrobials were locomotor disorders (58%) and respiratory diseases (37%). The remaining 5% was represented by other reasons such as gastrointestinal diseases, abscess, ear infection and horn fracture. Data on antimicrobial use according to the class and reason of administration are presented in Table 6.

NO-QUA and QUA groups did not differ significantly in terms of frequency of animals treated for locomotor disorders (e.g., lameness, interdigital dermatitis and interdigital phlegmon) and other diseases, whereas they differed significantly for respiratory diseases (Table 7).

### 3.4. Effects of Farm, Quarantine, and Season on Antimicrobial Use in Males

Antimicrobial use differed significantly between farms and groups. Indeed, NO-QUA group had higher TI100it than QUA group (3.76 vs. 3.46, respectively). In addition, males that arrived at the fattening farm during the coldest months of the year had higher TI100it compared with animals that arrived in spring and summer (Table 8).

## 4. Discussion

### 4.1. Effects of Farm, Quarantine and Season on Performance Traits

Farm was an important source of variation of performance traits during the fattening cycle in both male-rearing farms and female-rearing farms; in fact, different management strategies applied on-farm are crucial in determining growth performance [28]. Different feeding strategies supplied to the animals also play an essential role in explaining the variability of performance observed in our study. According to the general feeding management applied on specialized Italian beef fattening farms, females receive a diet with lower content of concentrates compared to males (Table 1) [26] to avoid an excessive fat deposition of female carcasses. This helps to explain why farmers are more likely to rear only one sex on their farms, i.e., to easily manage the different diets.

The strategy of quarantine showed a positive effect on performance of male-rearing farms. In the present study, beef cattle underwent a long travel from France to Italy (Veneto region) and a process of mixing both before travelling and at arrival to the fattening farms. Currin and Whittie [29] reported that transportation, especially during cross-country travels, is a stressful event for the animals, and Benavides et al. [30] reported higher likelihood of cross-contamination among animals for bovine viral diarrhea virus when farms shared transport vehicles or animals were transferred in contaminated vehicles. The aforesaid stressful events combined to the process of adaptation to a new farm environment can increase animals’ susceptibility to diseases (e.g., BRD and bovine viral diarrhea virus) which in turn may affect their performance [25,31,32]. Indeed, the BRD is one of the most detrimental health issues affecting beef cattle and it is usually associated with a general depression-like status and a decrease of appetite of the animals [33]. Thus, it is likely that the implementation of a 30-day period of quarantine before entering the standard fattening cycle, was an effective measure in reducing diseases in Charolaise males as suggested by an improved animal performance and the lower number of parenteral treatments administered for respiratory diseases. However, a positive effect of quarantine on performance traits was not observed in Charolaise female-rearing farms, likely due to their lower susceptibility to diseases, mainly BRD [34], as supported by the low number of females treated (30 out of 630 females) reported in this study. These findings are also in line with other studies reporting that male beef calves were at higher risk of diseases and had an increased likelihood to die due to BRD than females [35,36]. Another factor that may help to explain the higher susceptibility of males to respiratory diseases, can be the difference in the level of concentrates supplied to their diet [36,37,38]. For instance, Galyean et al. [37] reported a trend for increased BRD morbidity with increasing levels of concentrate, in particular when such level was above 50% of the total diet. The difference in diet composition between male-rearing farms and female-rearing farms is in line with data presented in literature. Indeed, the level of concentrates administered to males was higher than the amount provided to females and with a percentage above 60%. Finally, it is also likely that differences in animal management with regards to care/handling was applied on-farm [39], for instance due to the fact that females are more docile and easier to handle than males. Poor animal handling can lead to high levels of stress which in turn may impair the animals’ immune systems and increase their susceptibility to diseases [40].

The season of arrival to the fattening farm is as an important source of variation of animal’s performances [28]. In our study, males had better performances during spring, likely due to warmer temperatures and lower humidity typical of this period of the year in Veneto region, which may contribute to reduce the risk of BRD. However, both heat stress and exposure to diseases may lead to a reduction of animal performances, as observed by Sturaro et al. [28], who reported a reduced ADG in beef cattle with high temperatures experienced during summer.

### 4.2. Effects of Farm, Quarantine, and Season on Antimicrobial Use

Italian beef farming is mainly characterized by young animals imported from France and reared under intensive fattening conditions in the north-east of Italy. Although this system is recognized as a positive integration between the two countries in terms of exploitation of resources available [41], the long transport distance, the lack of a preventive vaccination program and stress exposure might increase the need of AMU. Specifically, the long transport distance that animals have to undergo is a common practice in beef industry [42], thus making the findings of our study applicable to other international beef fattening realities. In this study, 156 animals out of 1206 were treated at least once with a VMP. However, we also observed that females were less treated than males (30 vs. 126, respectively), in accordance with our previous studies carried out on a larger sample of beef fattening farms [31,43]. A different level of concentrates between male-rearing and female-rearing farms can also contribute to explain such a difference in the number of animals treated. In fact, according to the study of Fluharty and Loerchthe [38], higher percentage of concentrates led to a greater number of treatments required for sick beef calves. Penicillins was the most used class of antimicrobials, thus highlighting a wide exploitation of broad-spectrum antimicrobials which are known for their contribution to the development of AMR [44,45].

The main reason of AMU in the current study was for locomotor disorders followed by respiratory diseases. The Charolaise beef breed is known for its high BW compared to other breeds [26]. This could be one of the reasons that makes these animals more prone to develop lameness [46], a welfare issue that can increase under intensive conditions when there is a lack of an appropriate flooring system and the animals reach a final BW greater than 700 kg [47].

The percentage of parenteral treatments and the TI100it were lower in QUA than NO-QUA group suggesting that providing an initial 30-day period of quarantine to the animals arriving to the fattening farm led to a reduction of AMU. Indeed, biosecurity measures, such as the practice of quarantine, are essential to reduce the spread of diseases [16,19,25] and consequently may help to decrease the need of AMU as observed in the current study.

Significant differences for AMU among male-rearing farms may be explained by differences in feeding and management practices. Sharma et al. [11] reported the importance of a suitable diet in the prevention of diseases. Indeed, targeted feeding strategies are important to maintain appropriate animal health and welfare conditions and specifically they seem to be associated with the presence of locomotor disorders like lameness [46]. For instance, Compiani et al. [46] reported that a good optimization of the feed ration at the arrival to the farm combined to a gradual transition towards the ration of energy concentrates, helps to manage acidotic events and the associated risk of developing lameness in beef cattle. Therefore, although Charolaise beef breed requires a high level of starch and energy in the diet, a gradual transition to the new diet is essential to avoid locomotor impairments. This may help to clarify why Charolaise males-rearing farms were more likely to have animals more treated with antimicrobials for locomotor disorders.

A season effect on AMU was also reported in Charolaise male-rearing farms showing that animals arriving to the fattening farm in winter and autumn had higher TI100it compared to animals arriving in summer and spring. Similar results were reported in our previous study [31,47], where low temperatures and different humidity likely explained the greater likelihood of BRD observed in winter and autumn. Although Becker et al. [25] reported that there was no clear linkage between winter and AMU in young calves, we observed an increase of TI100it during the coldest months of the year.

According to the literature, factors such as transportation distance, farmer–veterinarian relationship, and variables associated to the pen such as the m^2^ can be considered as possible sources of variation of animal health, performance and AMU [48,49,50] whereby justifying their investigation through the statistical model. However, transportation data were not available, and the five farms involved in the study were managed by the same veterinarians equally trained and employed by the cooperative of beef producers (AZoVe). Thus, in this case we did not consider necessary to investigate the latest effect in the model. Instead, we tested the area (m^2^) of the pens, but we decided to remove this effect from the final model because not significant whereby confirming that differences between pens within and between farms of same sex were not significant. Future intervention studies to further investigate the effect of new strategies on AMU in beef farms are needed and inclusion of other farm characteristics may contribute to provide target and cost-effective information for a more holistic view.

## 5. Conclusions

Overall, this study showed that the implementation of the practice of quarantine was a feasible strategy to reduce AMU in beef production without compromising animal health and performances. Specifically, this strategy was effective in male-rearing farms and in the reduction of AMU administered for respiratory diseases. Building quarantine facilities ex-novo is a cost for beef farmers, but the reduction of AMU on the long term may help to compensate this initial economic investment. Moreover, stricter application of current EU policies that promote high standards of animal welfare may drive farmers towards the implementation of a more welfare-friendly farm helping them not only to apply a more judicious AMU in beef cattle but also to cover such initial costs. Findings of this study can be representative of similar beef fattening farm realities worldwide, specifically those characterized by animals reared at pasture in the first part of their life, followed by a long transportation distance to reach the fattening farm and by intensive fattening conditions. On-farm, a rich energy diet is supplied to the animals to reach the required final BW in relatively short time. Another aspect that can make our results easily exploitable is that the Charolaise breed used for the study is a cosmopolitan breed, thus making our findings applicable to other realities. Nevertheless, further research should investigate AMU in other breeds or multi-breed farms. In addition, it would be interesting to investigate the effect of quarantine on AMU in other types of beef fattening systems.

## Figures and Tables

**Table 1 animals-12-00116-t001:** Average characteristics of the diets provided to the animals according to their sex.

Diet Composition ^2^	Female-Rearing Farms ^1^	Male-Rearing Farms ^1^
Total ingestion, kg	16.9	16.5
DM, kg	9.0	9.9
ME, UFC	8.6	10.0
PDI, g	820.7	966.6
PDIN, g	796.2	890.4
Concentrates, %	54.2	63.7
Forages, %	45.8	36.3
Chemical Composition		
Moisture, %	45.8	39.9
CP, %	13.7	13.4
EE, %	4.2	3.9
CF, %	17.3	14.2
Ash, %	5.2	4.9
NDF, %	36.8	32.1
Starch, %	29.5	33.9

^1^ The values were calculated as an average of the diet administered by every farm involved in the trial per sex (2 male-rearing farms and 3 female-rearing farms). ^2^ DM = dry matter; ME = metabolizable energy; PDI = protein digestible in the intestine; PDIN = true protein absorbable in the intestine when *n* is limiting in the rumen; CP = crude protein; EE = ether extract; CF = crude fiber; NDF = neutral detergent fiber.

**Table 2 animals-12-00116-t002:** Descriptive statistics of performance traits ^1^ by sex of Charolaise cattle.

Sex	*n*	Trait	Mean	SD	Minimum	Maximum
Female	630	BW0, kg	320.6	20.0	267	380
		BW30, kg	366.9	27.5	260	443
		BWfinal, kg	559.6	41.4	405	712
		LFC, days	194.9	7.8	183	208
		ADG30, kg/day	1.47	0.54	−1.25	2.49
		ADGtot, kg/day	1.23	0.2	0.39	1.87
Male	576	BW0, kg	403.4	19.1	343.0	460.0
		BW30, kg	470.2	27.0	394	552
		BWfinal, kg	719.6	49.1	570	870
		LFC, days	191.2	5.4	117	207
		ADG30, kg/day	1.99	0.65	−1.17	3.19
		ADGtot, kg/day	1.65	0.24	0.74	2.65

^1^ BW0 = body weight at arrival to the fattening farm; BW30 = body weight 30 days after arrival to the fattening farm; BWfinal = body weight at the end of the fattening cycle; LFC = length of the fattening cycle; ADG30 = average daily gain 30 days after arrival to the fattening farm; ADGtot = average daily gain of the fattening cycle.

**Table 3 animals-12-00116-t003:** Least squares means (LSM) and standard error (SE) of performance traits ^1^ for farm, quarantine, and season of arrival effects in males of Charolaise breed (*n* = 576).

Effect	Category	BW30, kg			BWfinal, kg			ADG30, kg/day			ADGtot, kg/day		
		LSM	SE	*p*-Value	LSM	SE	*p*-Value	LSM	SE	*p*-Value	LSM	SE	*p*-Value
Farm	1	470.3 ^a^	1.45	0.0313	727.4 ^a^	2.93	0.0001	1.97 ^a^	0.04	0.4399	1.68 ^a^	0.02	0.0236
	2	465.4 ^b^	1.80		709.7 ^b^	3.68		1.93 ^a^	0.05		1.62 ^b^	0.02	
Group ^2^	NO-QUA	464.6 ^b^	1.68	0.0068	713.4 ^b^	3.42	0.0333	1.85 ^b^	0.05	0.0020	1.62 ^b^	0.02	0.0204
	QUA	471.1 ^a^	1.68		723.7 ^a^	3.44		2.05 ^a^	0.05		1.68 ^a^	0.02	
Season of arrival	Autumn	461.9 ^b^	2.32	<0.0001	705.3 ^b^	4.65	0.0001	1.65 ^c^	0.06	<0.0001	1.58 ^b^	0.02	<0.0001
	Winter	464.2 ^b^	3.48		727.0 ^b^	7.12		1.94 ^a,b,c^	0.09		1.69 ^a,b^	0.04	
	Spring	476.2 ^a^	1.86		729.3 ^a^	3.77		2.21 ^a^	0.05		1.72 ^a^	0.02	
	Summer	469.0 ^b^	1.60		712.5 ^b^	3.27		2.01 ^b^	0.04		1.61 ^b^	0.02	

^1^ BW30 = body weight 30 days after arrival to the fattening farm; BWfinal = body weight at the end of the fattening cycle; ADG30 = average daily gain 30 days after arrival to the fattening farm; ADGtot = average daily gain of the fattening cycle. ^2^ NO-QUA = animals not subjected to quarantine; QUA = animals subjected to quarantine. ^a,b,c^ Means with different superscript letters within trait and effect are significantly different according to Bonferroni’s adjustment (*p* < 0.05).

**Table 4 animals-12-00116-t004:** Least squares means (LSM) and standard error (SE) of performance traits ^1^ for farm, quarantine, and season of arrival effects in females of Charolaise breed (*n* = 630).

Effect	Category	BW30, kg			BWfinal, kg			ADG30, kg/day			ADGtot, kg/day		
		LSM	SE	*p*-Value	LSM	SE	*p*-Value	LSM	SE	*p*-Value	LSM	SE	*p*-Value
Farm	1	359.4 ^b^	1.39	<0.0001	565.6 ^a,b^	2.74	0.0026	1.31 ^b^	0.04	<0.0001	1.21 ^a^	0.01	0.0244
	2	372.2 ^a^	1.41		571.4 ^b^	3.28		1.57 ^a^	0.04		1.26 ^a^	0.02	
	3	370.2 ^a^	1.43		556.8 ^a^	2.78		1.56 ^a^	0.04		1.26 ^a^	0.01	
Group ^2^	NO-QUA	367.6 ^a^	1.16	0.6595	565.7 ^a^	2.41	0.5079	1.48 ^a^	0.03	0.9870	1.25 ^a^	0.01	0.5698
	QUA	366.9 ^a^	1.13		563.5 ^a^	2.36		1.48 ^a^	0.03		1.24 ^a^	0.01	
Season of arrival	Autumn	361.0 ^b^	1.60	<0.0001	556.4 ^b^	3.13	<0.0001	1.40 ^b^	0.05	0.0015	1.24 ^b^	0.02	<0.0001
	Winter	365.9 ^b^	2.11		589.2 ^a^	4.21		1.53 ^a,b^	0.06		1.33 ^a^	0.02	
	Spring	366.3 ^b^	1.58		561.0 ^b^	3.94		1.40 ^b^	0.05		1.20 ^b^	0.02	
	Summer	375.8 ^a^	1.37		551.8 ^b^	2.66		1.59 ^a^	0.04		1.21 ^b^	0.01	

^1^ BW30 = body weight 30 days after arrival to the fattening farm; BWfinal = body weight at the end of the fattening cycle; ADG30 = average daily gain 30 days after arrival to the fattening farm; ADGtot = average daily gain of the fattening cycle. ^2^ NO-QUA = animals not subjected to quarantine; QUA = animals subjected to quarantine. ^a,b^ Means with different superscript letters within trait and effect are significantly different according to Bonferroni’s adjustment (*p* < 0.05).

**Table 5 animals-12-00116-t005:** Number and percentage of treatments administered to the animals per class of antimicrobial and group ^1^ (QUA and NO-QUA) considering the number of active ingredients included in each parenteral treatment (*n* = 763).

Class of Antimicrobial	Total		QUA		NO-QUA	
	*n*	%	*n*	%	*n*	%
Penicillins	223	29.2	116	15.2	107	14.0
Amphenicols	150	19.7	56	7.3	94	12.3
Fluoroquinolones	120	15.7	45	5.9	75	9.8
Aminopenicillins	106	13.9	31	4.1	75	9.8
Penicillins (antistaphylococcal) ^2^	57	7.5	22	2.9	35	4.6
Sulfonamides	48	6.3	24	3.2	24	3.2
Tetracyclines	41	5.4	27	3.5	14	1.8
Aminoglycosides	11	1.4	3	0.4	8	1.1
Lincosamides	7	0.9	0	0.0	7	0.9
Total	763	100.0	324	42.5	439	57.6

^1^ NO-QUA = animals not subjected to quarantine; QUA = animals subjected to quarantine. ^2^ Beta-lactamase resistant penicillins (e.g., cloxacillin and dicloxacillin).

**Table 6 animals-12-00116-t006:** Number of treatments administered to the animals per class of antimicrobial and reason of administration (respiratory disease, locomotor disorder, other) considering the number of active ingredients included in each parenteral treatment (*n* = 763).

Class of Antimicrobial	Respiratory	Locomotor	Other	Total
	*n*	*n*	*n*	*n*
Penicillins	0	220	3	223
Amphenicols	141	0	9	150
Fluoroquinolones	45	73	2	120
Aminopenicillins	47	58	1	106
Penicillins (antistaphylococcal) ^1^	0	57	0	57
Sulfonamides	2	2	44	48
Tetracyclines	2	39	0	41
Aminoglycosides	11	0	0	11
Lincosamides	7	0	0	7
Total	255	449	59	763

^1^ Beta-lactamase resistant penicillins (e.g., cloxacillin and dicloxacillin).

**Table 7 animals-12-00116-t007:** Number of parenteral treatments with antimicrobials administered to the animals included in the study (*n* = 1206) according to the reason of administration and group.

Group ^1^	Locomotor	Respiratory	Other	Total
NO-QUA	197 ^a^	165 ^a^	23 ^a^	385 ^a^
QUA	194 ^a^	82 ^b^	14 ^a^	290 ^b^

^1^ NO-QUA = animals not subjected to quarantine; QUA = animals subjected to quarantine. ^a,b^ Different superscript letters within reason of administration indicate significant differences between NO-QUA and QUA groups according to Chi-Square test (*p* < 0.05).

**Table 8 animals-12-00116-t008:** Least squares means (LSM) and standard error (SE) of treatment incidence 100 for Italy (TI100it) for farm, group, and season of arrival effects in males of Charolaise breed (*n* = 576).

Effect	Category	TI100it		
		LSM	SE	*p*-Value
Farm	1	3.39 ^b^	0.10	0.006
	2	3.84 ^a^	0.14	
Group ^1^	NO-QUA	3.76 ^a^	0.12	0.033
	QUA	3.46 ^b^	0.10	
Season of arrival	Autumn	4.13 ^a^	0.20	0.002
	Winter	3.67 ^a,b^	0.25	
	Spring	3.38 ^b^	0.12	
	Summer	3.31 ^b^	0.10	

^1^ NO-QUA = animals not subjected to quarantine; QUA = animals subjected to quarantine. ^a,b^ Means with different superscript letters within effect are significantly different according to Tukey–Kramer adjustment (*p* < 0.05).

## Data Availability

The data presented in this study are available on request from the corresponding author. The data are not publicly available due to their sensitive nature.

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
