# Peer review of "Promoting Judicious Antimicrobial Use in Beef Production: The Role of Quarantine"

_animals, 2022, doi:10.3390/ani12010116_

Round 1
Reviewer 1 Report
Article review:
Manuscript ID: animals-1487932
Title: Promoting judicious antimicrobial use in beef production: the role of quarantine
Comments and Suggestions for Authors:
Methodologically well structured article with very relevant results. It should be better discussed
why the differences between groups (QUA and NO-QUA) were significant only for males.
Minor Concerns:
2. Materials and Methods
It would be important in this chapter to describe how regularly the animals were weighed to
determine the average daily gain (ADG) 30 and tot.
I suggest removing the following sentence: “Data were tested for normality by checking
skewness and kurtosis and visual inspection of the normal plot”.
3. Results: Table 2: on page 6:
- It is recommended to check that the maximum values of ADG30 and ADGtot parameters of
2.49 and 1.87 in females and 3.19 and 2.65 in males are correct, because they are very high.
Author Response
Reviewers’ comments are listed below, and authors’ responses (AU) are shown beneath each comment. Changes in the marked revised manuscript are highlighted in yellow.
AU: We would like to thank the Editor and the Reviewers for their professional help in reviewing the manuscript.
Reviewer 1
Comments and Suggestions for Authors:
Methodologically well-structured article with very relevant results. It should be better discussed why the differences between groups (QUA and NO-QUA) were significant only for males.
AU: We thank the reviewer for this comment. Therefore, more details were added to the discussion to explain why significant differences between QUA and NO-QUA groups were reported only for male-rearing farms (See lines 349-363 and lines 382-385).
Minor Concerns:
Materials and Methods
It would be important in this chapter to describe how regularly the animals were weighed to determine the average daily gain (ADG) 30 and tot.
AU: This information was further elaborated and it is now available in the manuscript at lines 154-159. In particular, the ADG30 was calculated as the ratio between the weight gained (i.e. the difference between body weight at 30 days after the start of the fattening cycle and the body weight at arrival to the farm) and the number of days since the last weight (30 days). The ADGtot was calculated as the ratio between the weight gained (i.e. the difference between body weight prior to reach the slaughterhouse and the body weight at arrival to the farm) and the number of days spent to reach slaughter (i.e. date of slaughter – date of start of the fattening cycle).
I suggest removing the following sentence: “Data were tested for normality by checking skewness and kurtosis and visual inspection of the normal plot”.
AU: We thank the reviewer for the comment and we agreed to make the sentence shorter as follow ‘Data were tested for normality’. Indeed, checking the data for normality is an essential step in order to explore the data and set the right model for the statistical analysis (line 192).
Results:
Table 2 on page 6: It is recommended to check that the maximum values of ADG30 and ADGtot parameters of 2.49 and 1.87 in females and 3.19 and 2.65 in males are correct, because they are very high.
AU: We thank the reviewer for this comment and we confirm that data are correct. Regarding the maximum values of ADG, they are due to very few animals that we decided to maintain for analysis because the overall mean ADGs were still biologically relevant and consistent with the literature and our previous studies; thus, also avoiding to discard potential treated animals. Nevertheless, it is likely that the high maximum values reported for these few animals can be explained with the time of weighing the animals and their feeding time. At this stage, the animals can eat c. 15-17 kg of unifeed per day, and these animals may have been weighed while still performing rumination, thus likely clarifying the maximum values obtained for both ADGs.
Reviewer 2 Report
This study clarified the prudent use of antibacterial agents with sufficient data and scientific evidence. The data are potentially interesting and worthy of eventual publication. Thank you for your interesting research.
Author Response
Reviewers’ comments are listed below, and authors’ responses (AU) are shown beneath each comment. Changes in the marked revised manuscript are highlighted in yellow.
AU: We would like to thank the Editor and the Reviewers for their professional help in reviewing the manuscript.
Reviewer 2
This study clarified the prudent use of antibacterial agents with sufficient data and scientific evidence. The data are potentially interesting and worthy of eventual publication. Thank you for your interesting research.
AU: We would like to thank the reviewer for his/her feedback. We appreciate knowing that our work will add new notions to the literature on potential cost-effective strategies to reduce the use of antimicrobials in beef farming.
Reviewer 3 Report
Limitations of present manuscript
- Health background of animals (vaccintaions, antiparasitic and /or Vit Complex administration etc). This may be a bias particulraly when no randomazation is applied
- the effect of season and consequently enviroment is key factor for respiratory problems. It should be further analyzed if there is a potential significant regression between weather and heath risk
- Who was examining animals? Was he blind to the allocation of groups?
Author Response
Reviewers’ comments are listed below, and authors’ responses (AU) are shown beneath each comment. Changes in the marked revised manuscript are highlighted in yellow.
AU: We would like to thank the Editor and the Reviewers for their professional help in reviewing the manuscript.
Reviewer 3
Health background of animals (vaccintaions, antiparasitic and/or Vit Complex administration etc). This may be a bias particulraly when no randomazation is applied.
AU: We agree with the reviewer that this information was missing, thus it was added to the text (See lines 133-136). We clarified that the experimental animals went through the same health protocol. Specifically, they did not receive any vaccinations nor antimicrobial treatments in France. Whereas, at their arrival to the Italian farms involved in the study, the same vaccination programme (i.e. polyvalent vaccine for BRD) and parasitic control programme were administered to the animals.
The effect of season and consequently enviroment is key factor for respiratory problems. It should be further analyzed if there is a potential significant regression between weather and heath risk.
AU: We agree with the reviewer that the environment is a key factor for respiratory diseases, however the aim of this study was to investigate potential cost-effective biosecurity strategies, such as the practice of quarantine, that can be easily applied on-farm in order to reduce antimicrobial use (AMU) in beef production. Indeed, the impact of weather conditions on the incidence of respiratory diseases has been already explored in literature (e.g. Cusack et al., 2007 - https://doi.org/10.1111/j.1751-0813.2007.00184.x; Gay & Barnouin, 2009 - https://doi.org/10.1016/j.prevetmed.2009.02.013), while little is known about the strategies that can contribute to the reduction of AMU without compromising animals’ performance in beef production. We decided to include the season effect in the model because according to our previous study, where we investigated potential sources of variation of AMU in beef production, an increase of AMU during the coldest months of the year was reported, likely due to a higher incidence of respiratory diseases (Diana et al., 2021 - https://doi.org/10.1016/j.animal.2020.100091). In addition, even if the farms involved in the study, all located in the same regional area (i.e. Veneto), had similar weather conditions, data about temperature and humidity on-farm were not available. Therefore, an in-depth investigation of the link between temperature, humidity and health risk was difficult to achieve.
Who was examining animals? Was he blind to the allocation of groups?
AU: The animals included in the study were examined by the same veterinarians who are equally trained and work for the cooperative of beef producers (AZoVe) where the farms belong to. They provide medical support/advice to all the farms associated with this cooperative and they are not aware on whether the farms are or are not involved in research trials. This helps to avoid any potential bias related to the examination of the animals and related administration of antimicrobials. More details were added to the manuscript (See lines 172-175).
Round 2
Reviewer 3 Report
My suggestions were satisfied